# A Dynamic Model for the Study and Simulation of the Pantograph–Rigid Catenary Interaction with an Overlapping Span

Jesús Benet [1], Fernando Cuartero [2], Tomás Rojo [2], Pedro Tendero [2] and Enrique Arias [2,*]

[1] Departamento Mecánica Aplicada e Ingeniería de Proyectos, Universidad de Castilla-La Mancha, 02071 Albacete, Spain; jesus.benet@uclm.es
[2] Departamento de Sistemas Informáticos, Universidad de Castilla-La Mancha, 02071 Albacete, Spain; Fernando.Cuartero@uclm.es (F.C.); Tomas.Rojo@uclm.es (T.R.); Pedro.Tendero@uclm.es (P.T.)
[*] Correspondence: enrique.arias@uclm.es; Tel.: +34-967599200 (ext. 2497)

**Abstract:** In this paper, the authors present a mathematical and engineering model to optimally calculate the dynamic equation on the pantograph–catenary interaction when considering a rigid catenary with an overlapping span. The model starts from well-known methods adapted to the special features of rigid catenary. As a result, an algorithm for the integration of a dynamic equation based on explicit methods is provided. Moreover, from this algorithm, a reliable, efficient, and user-friendly software tool called RICATI is developed in order to approach the model to railway-based companies. The results show the usefulness of an application. such as RICATI, to check the behavior of the configuration initially established for a catenary, allowing solutions to be obtained for the problems encountered when simulating the passage of the pantograph (or pantographs), not only for the overlapping span but also for the entire catenary. That encourages us to continue future works.

**Keywords:** transportation; pantograph/catenary interaction; rigid catenary; software tool; infrastructure engineering; information technology





## 1. Introduction

Rail transportation is fundamentally important in modern societies. As a result, the development of railway technologies that allow the efficient circulation of people and goods faces challenges related to a variety of technical problems. Addressing these challenges is the object of study in research centers and universities. Throughout the history of the railway, different types of traction have been proposed for the tractor units. Traction by electric power is currently the most widely used system, in which the power is mainly supplied via an overhead contact line or catenary, and the pantograph is the mechanism of the tractor unit used to capture this energy.

Two types of catenaries can be considered. Flexible catenary, or elastic catenary, is an extensional component (bending is almost negligible but has to be considered). On the other hand, a rigid catenary is a bending component (axial extension is negligible). The study of the pantograph–catenary dynamic interaction to achieve optimum operating conditions for railways is therefore a fundamental technological problem. For this purpose, the contact force must be as uniform as possible. Contact losses or take-offs must be avoided, for which it is necessary to establish optimal assembly parameters. Thus, there is significant research interest in the development of a mathematical model that allows realistic simulations, and that is also computationally efficient.

The problem of pantograph–catenary dynamic interaction is highly important when the pantograph interacts with the elastic catenary, which has been the subject of a wide range of publications (see references [1–18]). This number of studies on flexible catenary is due to the wide use of this type of catenary in high-speed railways that require higher

performance. However, this issue also has special relevance when the interaction occurs with a rigid catenary, as is the case in tunnels and underground roads; however, few scientific studies have been conducted for this problem. In the case of this power system, the element that transmits the electrical energy is not a wire but a rigid rail. Due to its weight, this rigid rail cannot be maintained parallel to the track using tension or by suspending it from another wire with an amount of deflection; rather, it is necessary to increase the number of supports on which it must be suspended to decrease the distance between them. For example, a typical span of a rigid catenary has a length of about 10–12 m, whereas the span for the flexible catenary is around 50–60 m.

It is well known that, in the cases of rigid catenary, the speed of the units is restricted to maximum values of around 100–120 km/h. A small number of studies have been published for this type of line (see references [19–21]). Thus, it is an important goal of scientific research to raise this maximum speed limit.

In addition, to achieve greater rigidity of the catenary, which may be a cause of a significant fluctuation of the contact force, the sudden variation of this force must be added when the pantograph circulates through the spans of transition between two series of spans. These variations are more frequent because the length of the spans is notably less than in the case of flexible catenaries. No publications in the scientific literature address this problem in a satisfactory and detailed manner. In the case of overlapping spans, it is also necessary to configure the beams with a special slope-shaped geometry that ensures that the flow of the pantograph is as smooth as possible between the sequences of spans.

In this study, we addressed the problem of dynamic pantograph–rigid catenary interaction with an overlapping span. This problem has significant practical importance and, as previously indicated, has not been closely studied in the scientific literature. For this purpose, we present a methodology based on the combination of well-known mathematical models. Based on this model, a computer application for the study and simulation of the dynamic interaction of the pantograph with a rigid catenary is developed. This model is applied to two catenary series with an overlapping span and several pantographs in circulation. In the first section of this paper, the mathematical foundations of the model are explained and the algorithm of the developed computer program is presented. In the subsequent section, the computational aspects and results of the simulations are discussed.

The main novelty of this work lies in the application of well-known methods to study the dynamic interaction between the pantograph and rigid catenary, and the establishment of a development methodology. Therefore, the methods used in this paper are not novel. However, the application of all of them together to solve the problem of pantograph–rigid catenary interaction is a novelty. Much of the work found in the literature has been developed for flexible catenary. However, in this paper, existing methods and algorithms have been adapted in order to solve the case of pantograph–rigid catenary interaction. It should be noted that the mechanical characteristics of a rigid catenary are different from those of a flexible catenary. As a result, a specific study is necessary for this type of catenary because it is not feasible to consider a flexible catenary with extreme rigidity

In addition, a software tool is introduced that addresses the problem of pantograph–rigid catenary dynamic interaction practically and efficiently from the perspective of both the accuracy of the results and the response time.

To sum up, these results for rigid catenary, as well as the underlying application, constitute the novelty of this method.

## 2. Materials and Methods

### 2.1. Dynamic Equations of the Catenary Pantograph System

The pantograph–catenary system is initially composed of two subsystems interacting with each other under restricted conditions. The dynamic equations for a time instant $t_n$ are shown in Equation (1), where $M$ is the mass matrix, assumed to be constant; $C_n$ is the damping matrix; $K_n$ is the stiffness matrix; $\varphi_n$ is the matrix of the constraint conditions; $R_n$ is the vector of external loads on the system; $q_n$ is the vector of coordinates that define the

position of the system; and $\lambda_n$ is the vector corresponding to the restriction forces. All of the latter elements are assumed to be variables, either in totality or in some of their terms for each instant $t_n$. This results in a non-linear system of differential equations of second order whose integration in time will yield the evolution of the coordinates $q_n$ and contact forces $\lambda_n$. The subscript $n$ represents the value of variables at the instant $t_n$.

$$\begin{pmatrix} M & 0 \\ 0 & 0 \end{pmatrix} \begin{pmatrix} \ddot{q}_n \\ \ddot{\lambda}_n \end{pmatrix} + \begin{pmatrix} C_n & 0 \\ 0 & 0 \end{pmatrix} \begin{pmatrix} \dot{q}_n \\ \dot{\lambda}_n \end{pmatrix} + \begin{pmatrix} K_n & \varnothing_n^t \\ \varnothing_n & 0 \end{pmatrix} \begin{pmatrix} q_n \\ \lambda_n \end{pmatrix} = \begin{pmatrix} R_n \\ 0 \end{pmatrix} \quad (1)$$

When considering two series of spans, as shown in Figure 1, it is necessary to partition the system coordinates and the different elements of the equation. In the case of two pantographs, as represented in Figure 1, this results in the following structure:

$$q_n = \begin{pmatrix} q_{c1} \\ q_{c2} \\ q_{p1} \\ q_{p2} \end{pmatrix}, \quad \lambda_n = \begin{pmatrix} \dot{\lambda}_1 \\ \dot{\lambda}_2 \end{pmatrix}, \quad K_n = \begin{pmatrix} K_{c1} & 0 & 0 & 0 \\ 0 & K_{c2} & 0 & 0 \\ 0 & 0 & K_{p1} & 0 \\ 0 & 0 & 0 & K_{p2} \end{pmatrix}, \quad R_n = \begin{pmatrix} R_{c1} \\ R_{c2} \\ R_{p1} \\ R_{p2} \end{pmatrix} \quad (2)$$

where $q_{c1}$, $q_{c2}$, $q_{p1}$, and $q_{p2}$, represent, respectively, the coordinates of the two series of spans (subscript $c$ for catenary) and the two pantographs (subscript $p$). A similar notation may be used for the stiffness matrix and the other terms of Equation (1). The mass and damping matrices will have a structure similar to that of the stiffness matrix according to Equation (2). The vector $\lambda_n$ of the restraining forces or contact forces is divided into terms corresponding to the internal forces between each of the pantographs and catenaries of the two series of spans. The subscript $n$ represents the value of variables at an instant $t_n$.

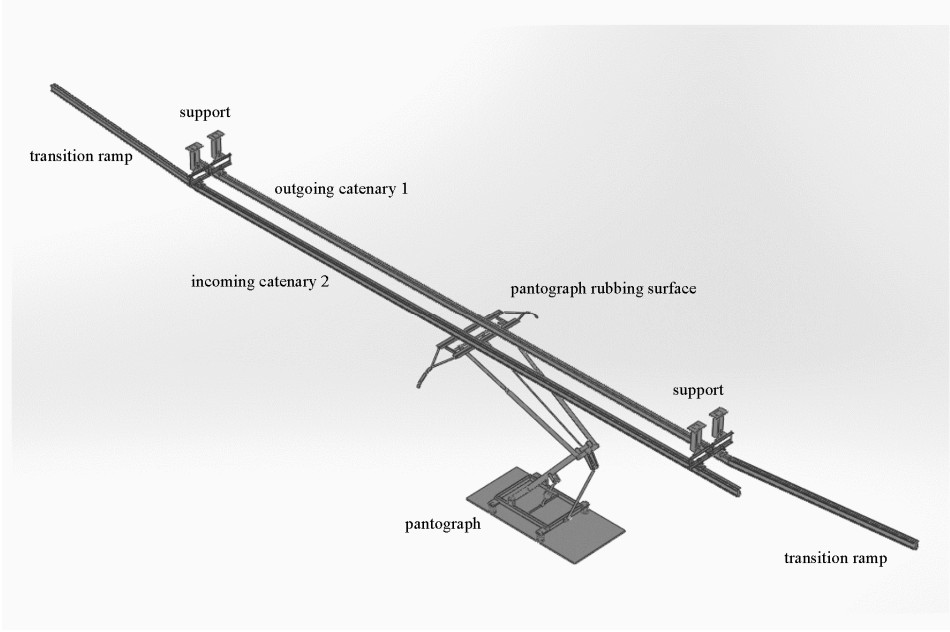

**Figure 1.** Interaction of one pantograph with two catenaries.

### 2.2. Beam Model for Rigid Catenary

In this work, the rigid catenary is modeled as a flexible beam. To derive a model for the catenary, the finite element method is used, as explained in references [22,23]. As the flexible beam is a simpler structure, the stiffness matrix is a band matrix with fewer elements which has a positive impact, not only on the computational cost in terms of calculation, but also in terms of storage, as it is treated as a sparse matrix.

The total length of the catenary is divided into a series of segments that behave as beams with flexural rigidity, according to the Euler–Bernoulli equation. In the discretization,

each node presents two coordinates corresponding to the displacement and rotation, resulting in matrices of 4 × 4 for each element, according to Figure 2.

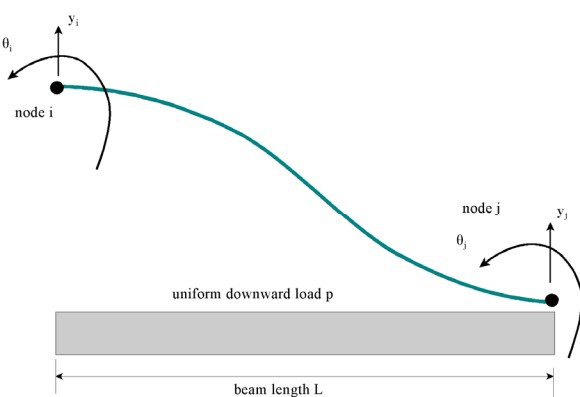

**Figure 2.** Beam element with bending stiffness.

A punctual mass matrix is assumed for the masses. Thus, the components of the coordinates and matrices for the beam element are:

$$q_c = \begin{pmatrix} y_i \\ \theta_i \\ y_j \\ \theta_j \end{pmatrix}, \quad M_c = \frac{m}{2} \begin{pmatrix} 1 & 0 & 0 & 0 \\ 0 & \frac{L^2}{12} & 0 & 0 \\ 0 & 0 & 1 & 0 \\ 0 & 0 & 0 & \frac{L^2}{12} \end{pmatrix}, \quad K_c = \frac{EI}{L^3} \begin{pmatrix} 12 & 6L & -12 & 6L \\ 6L & 4L^2 & -6L & -2L^2 \\ -12 & -6L & 12 & -6L \\ 6L & 2L^2 & -6L & 4L^2 \end{pmatrix}, \quad R_c = \begin{pmatrix} -\frac{pL}{2} \\ -\frac{pL^2}{12} \\ -\frac{pL}{2} \\ \frac{pL^2}{12} \end{pmatrix} \quad (3)$$

where $y_i$ and $\theta_i$ represent the displacement and rotation in the node $i$, respectively; $L$ represents the length of the element; $m$ represents its mass; $p$ is the weight of the beam per unit of length; $E$ is the elastic modulus of the beam material; and $I$ is the inertia moment of the section. These matrices are repeated for the different elements of the two catenaries. It can be seen that the different terms of the mass and stiffness matrices, and the vector of external loads, are constant in this case. For the catenary damping matrix, Rayleigh damping is assumed according to references [22,23], wherein the said matrix is a linear combination of the stiffness and mass matrix, per the equation:

$$C_c = \alpha K_c + \beta M_c \quad (4)$$

where $\alpha$ and $\beta$ are two constants that are determined from two assumed values of the critical damping for two different frequencies of interest. In our case, for 1 Hz, a critical damping percentage of 0.5% is assumed, and for 15 Hz, the assumed percentage is 1%. These values are obtained experimentally by trial and error. The experiment consists of the following. A video recording is made on two points along the line and the movement of the two points after passing the pantograph is recorded. Then, two points on the line coinciding with the real points are considered in the simulation program, and for two frequencies of interest (1 Hz and 15 Hz in this case) various damping percentages are assumed. The results of the simulation are compared with those obtained in the recording; it is then verified that the percentages given in the article are the ones that most closely match the real results.

### 2.3. Properties of the Rigid Catenary Section and the Supports

Figure 3 shows the details and geometric characteristics of a section of Furrer rigid beam, which is used in Spanish railways. The figure shows a normal section and a section with a joint flange. It can be seen that the normal beam is composed of two elements: the body of the beam itself, which is made of aluminum; and the contact wire, made of copper, through which the electric energy flows [20,21]. Obviously, Figure 3 consists of two elements: an aluminum beam and a copper wire. As we have two materials, we

should have two moments of inertia and two elastic moduli. However, according to [24], it is possible to obtain an equivalent single beam representing the composite beam. This approach is considered in this paper in order to simplify the model.

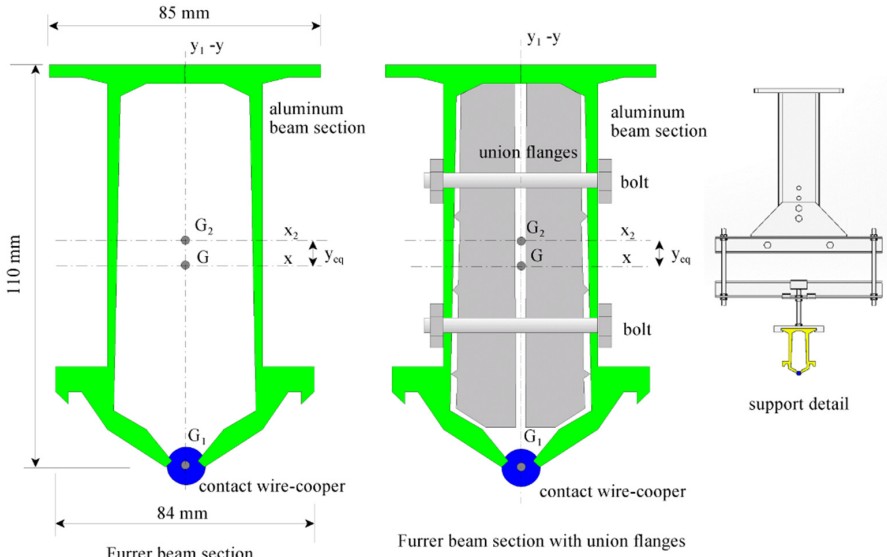

**Figure 3.** Details of the Furrer-type section and the fastening system.

As in the treatment by finite elements, according to Equation (3), we assume that the beam is composed of a single material. It is necessary to determine the equivalent values for the section of the area, the position of the center of gravity, the moment of inertia, and the density, as if this section is formed by a single material. Thus, assuming that the entire section is exclusively aluminum, we first define the relationship between the elastic modulus or Young's modulus, where the subscript 1 corresponds to copper, and subscript 2 to aluminum. This notation is also adopted for the remainder of the properties, and thus we have:

$$m = \frac{E_1}{E_2} \tag{5}$$

In Equation (5), $m$ describes the relationship between the moduli. To simplify the notation, the subscripts are excluded.

Equation (6) shows the area of the equivalent section $A_{eq}$; the new position of the center of gravity of section $G$ with respect to the position of the center of gravity of the aluminum section $G_2$ $y_{eq}$; the moment of inertia of the equivalent section with respect to the $x$-axis that passes through $G$ $I_{xeq}$; and the density of the equivalent section:

$$A_{eq} = A_2 + mA_1, \quad y_{eq} = \frac{mA_1y_1}{A_2 + mA_1}, \quad I_{xeq} = I_{x2} + mI_{x1}, \quad \rho_{eq} = \frac{\rho_1 A_1 + \rho_2 A_2}{mA_1 + A_2} \tag{6}$$

The details of the section of the rigid catenary section are shown in Figure 3, and the values for the Furrer beam are taken from reference [20,21]. After applying Equations (5) and (6) for the materials and sections of the beam, respectively, moments of inertia for the normal section of $I_{xeq}$ = 4,228,994 mm$^4$ and the flanged section of $I_{xeq}$ = 5,421,787 mm$^4$ are obtained. More details can be found in reference [20]. In addition, the spans of the beam are delimited by the supports, the detail of which is shown in Figure 3. The effect of the supports on the dynamic equations is assumed to be equivalent to that of a linear spring with stiffness $k_s = 1 \times 10^5$ N/m.

### 2.4. Pantograph Model

The pantograph is an articulated system whose purpose is to capture the electrical energy circulating through the contact wire and transmit it to the traction unit. To take into

account its effect on dynamic equations, it is usually modeled as a set of masses, springs, and shock absorbers, on which different forces can act: pushing force, aerodynamic force, friction, etc. The manufacturer usually provides the values of these parameters. Figure 4 shows a drawing of a pantograph interacting with two rigid catenaries and some models of point masses. In the proposed simulation, a pantograph of a single head mass is assumed, whose properties correspond to those in the European standard EN50318 (see reference [25]), as shown in Table 1.

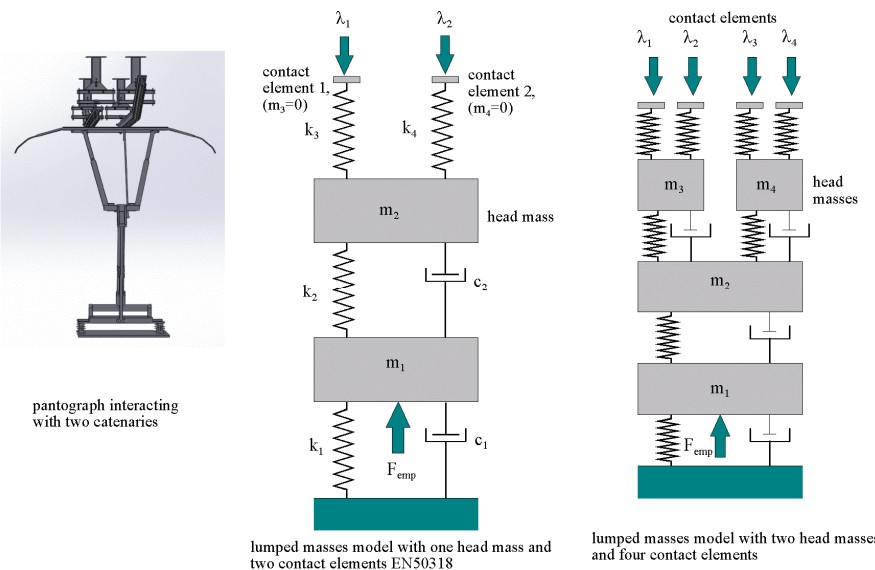

**Figure 4.** Pantograph models of punctual masses.

**Table 1.** Pantograph parameters according to EN5038.

|  | Effective Mass (kg) | Rigidity (N/m) | Cushioning (Ns/m) |
|---|---|---|---|
| Contact element | - | $k_3 = k_4 = k_{cont} = 50{,}000$ | - |
| Head collector | $m_2 = 7.2$ | $k_2 = 4200$ | $c_2 = 10$ |
| Base element | $m_1 = 15$ | $k_1 = 50$ | $c_1 = 90$ |
| Thrust force applied to the base: $F_{emp} = 120$ N | | | |

According to the cited EN50318 standard, to establish the contact model with the catenary, it is necessary to add a contact element of zero mass with a spring to the head mass of the collector $m_2$. Furthermore, because the pantograph can interact with two catenaries in our case, we assumed two contact elements with their corresponding stiffening springs, initially $k_3 = k_4 = k_{cont}$, although these values can be canceled out when take-off occurs, as explained below. In addition, it is assumed that each mass has a coordinate corresponding to its vertical displacement, resulting in four coordinates, including the contact elements, for the standard model. In the case in which the pantograph has two head masses, two contact elements must be added to each mass with the corresponding springs, according to Figure 4. Based on the above, the matrices of mass, rigidity, and the vector of external forces corresponding to the point mass model of the norm are:

$$M_p = \begin{pmatrix} m_1 & 0 & 0 & 0 \\ 0 & m_2 & 0 & 0 \\ 0 & 0 & 0 & 0 \\ 0 & 0 & 0 & 0 \end{pmatrix}, \quad K_p = \begin{pmatrix} k_1 + k_2 & -k_2 & 0 & 0 \\ -k_2 & k_2 + k_3 + k_4 & -k_3 & -k_4 \\ 0 & -k_3 & k_3 & 0 \\ 0 & -k_4 & 0 & k_4 \end{pmatrix}, \quad R_p = \begin{pmatrix} F_{emp} \\ 0 \\ 0 \\ 0 \end{pmatrix} \quad (7)$$

The damping matrix has a structure similar to that of the stiffness matrix. It should also be noted that the terms of the stiffness matrix are not constant because, when take-off

occurs, the contact disappears and the stiffness of the contact element $k_3$ or $k_4$ is canceled out. Furthermore, in the pantograph path, there are intervals in which the pantograph interacts only with a single catenary. In contrast, when the pantograph crosses the common transition zone, it interacts with the two catenaries, which also modify the structure of the stiffness matrix, as explained in later sections. In addition, in the data provided by the manufacturers, non-linearities can appear due to variations in the stiffness and damping, or the presence of friction forces. The mass matrix is always constant. Notice that the mass matrix is rank-deficient, meaning it is not possible to calculate its inverse. In this case, only the inverse of non-zero diagonal elements is calculated. More explanations about the calculation of the inverse of mass matrix will be given in Section 2.7.

### 2.5. Contact Pantograph/Catenary Model

The systems of pantograph and catenary are dependent but interact with each other under constraint conditions. In particular, the pantograph contact element slides on the contact wire of the catenary. Thus, a relationship between the positions is established with the pantograph contact element, represented by the contact element, and the position of the nodes of the catenary corresponding to the discretization using finite elements. In addition, this relationship varies depending on the time as the pantograph advances. The relationship between the coordinates of the catenary and the contact element of the pantograph corresponds to the term $\varphi_n$ that appears in the dynamic Equation (1). We suppose the distribution of the contact using a law that facilitates the sliding of the pantograph contact element along with the nodes of the catenary without any abruptness, as represented in Figure 5. This situation is also equivalent to assuming that the contact force is not punctual, but it is distributed over the surface of the pantograph contact element, according to the supposed distribution law (see Figure 5).

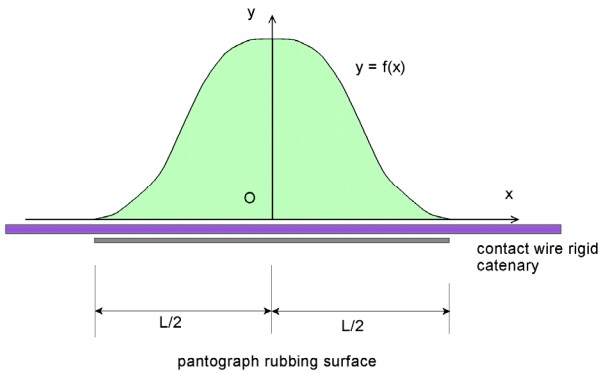

**Figure 5.** Law of pantograph–catenary contact distribution.

For convenience, an $O_{xy}$ axis system is assumed which moves with the pantograph, centered on the surface of the pantograph contact element. The variable $x$ represents the position of the different points of the catenary along the horizontal axis, in contact with the plate. If $L$ is the length of the friction surface, $y_3$ is the coordinate corresponding to the vertical position of the pantograph contact element, and $y_c(x)$ is the vertical position of the different points of the catenary, then $y = f(x)$, the function of contact distribution, will be fulfilled as follows:

$$\int_{-\infty}^{\infty} f(x)(y_c(x) - y_3)dx = 0 \tag{8}$$

with the condition:

$$\int_{-\infty}^{\infty} f(x)dx = 1 \tag{9}$$

Equations (8) and (9) allow the position of the contact element of the pantograph to be obtained, when the configuration of the catenary is known at time $t_n$, as a weighted average

of the position of the nodes of the catenary located on the surface of rubbing, according to the expression:

$$(y_3)_n = \int_{-\infty}^{\infty} f(x) y_c(x) dx \tag{10}$$

This results in a similar equation for $(y_4)_n$. For $f(x)$, we assumed a polynomial function defined as follows:

$$f(x) = \begin{cases} -\frac{32}{L^4}x^3 - \frac{24}{L^3}x^2 + \frac{2}{L}, & -\frac{L}{2} \leq x \leq 0 \\ \frac{32}{L^4}x^3 - \frac{24}{L^3}x^2 + \frac{2}{L}, & 0 \leq x \leq \frac{L}{2} \\ f(x) = 0, \ x < -\frac{L}{2} \cup x > \frac{L}{2} \end{cases} \tag{11}$$

The vertical position of any point of the catenary on the rubbing surface can be established from the position of the discretization nodes and the interpolation functions according to the finite element method, finally yielding the following relationship:

$$\varnothing_n q_n = 0 \tag{12}$$

### 2.6. Formulation of Restriction Conditions

Equation (12) should be repeated for all system restrictions, corresponding to all pantograph–catenary contacts. The number of restrictions will depend on the number of possible contacts. Thus, for the pantograph with one head mass, two possible contacts exist because the pantograph can interact with up to two catenaries when crossing the common contact zone.

If the pantograph has two head masses, there will be four possible contacts, and if two pantographs are considered in the simulation, the restriction conditions must be repeated for the second pantograph. Thus, Equation (12) is a matrix equation whose number of rows corresponds to the number of restriction conditions and whose number of columns corresponds to the number of degrees of freedom.

To simplify the model, we assume a single pantograph with a single head mass. Because there can be up to two contacts, the matrix of constraint conditions expressed for a moment $t_n$ has the form:

$$\varnothing_n = \begin{pmatrix} \varnothing_{1n} \\ \varnothing_{2n} \end{pmatrix} \tag{13}$$

where the row vector $\varphi_{1n}$ represents the restriction conditions of contact element 1 on catenary 1, and $\varphi_{2n}$ represents the restriction conditions of contact element 2 on catenary 2, according to Figure 5. In the formulation of Equation (13), the following cases are considered, as shown in Figure 6.

1.  The pantograph circulates through catenary 1 without reaching the common transition zone of the two catenaries, defined from the overlapping spans. In this case, interaction only exists with catenary 1 through contact element 1, and contact element 1 is activated and contact element 2 is deactivated.
2.  The pantograph circulates through catenary 2, having exceeded the common transition zone. In this case, there will only be interaction with catenary 2 through contact element 2; thus, contact element 2 is activated and contact element 1 is deactivated.
3.  The pantograph circulates in the common transition zone between the two catenaries. In this case, the pantograph interacts with the two catenaries, and the two contact elements are activated.

In the first case, the vector $\varphi_{1n}$ is calculated according to Equation (8), whereas the position of contact element 1, $(y_3)_n$, is calculated according to Equation (10). Because contact element 2 is deactivated, the restriction condition $\varphi_{2n}$ is formulated by requiring that the position of contact element 2, $(y_4)_n$, and the position of the head mass, $(y_2)_n$, at time $t_n$, be coincident; that is:

$$(y_2)_n - (y_4)_n = 0 \tag{14}$$

The second case is similar to the first, but with alternating conditions: $\varphi_{1n}$ is obtained by imposing the condition that the position of contact element 1, $(y_3)_n$, and the position of the head mass, $(y_2)_n$, at instant $t_n$ are coincident:

$$(y_2)_n - (y_3)_n = 0 \tag{15}$$

Matrix $\varphi_{2n}$ and the position of contact element 2, $(y_4)_n$, are calculated with equations similar to Equations (8) and (10). Finally, it must be taken into account that, regardless of the catenary with which the interaction occurs, the contact force can be canceled out because of a take-off or can appear suddenly after a take-off when a new coupling occurs. These circumstances must be considered when integrating Equation (1), as explained in the following section.

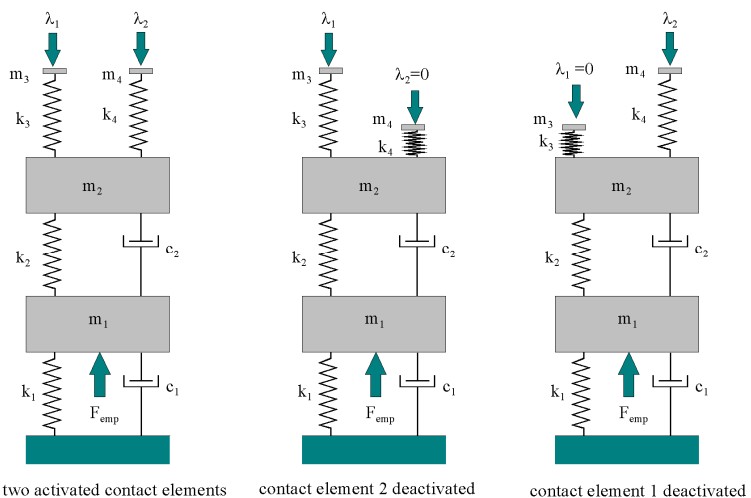

two activated contact elements      contact element 2 deactivated      contact element 1 deactivated

**Figure 6.** Pantograph with different activation cases in the contact elements.

### 2.7. Integration of Dynamic Equations

To proceed with the integration of the differential Equation (1), we applied the explicit method of the central differences. Explicit methods are more suitable than implicit methods for dealing with non-linearities, in the particular case of the pantograph–catenary interaction [23], because they correctly take into account the variation of the stiffness matrix. Non-linearities occur due to debonding and coupling. Due to the obvious stability problems of explicit methods, the integration step must be carefully adjusted, which is undertaken in this development. The critical integration step that conditions the stability can be estimated as the element length divided by the wave propagation speed, as shown in Equations (13.10)–(16) (p. 401) of [23]. Regarding the element length, in this work all elements have the same length. If this is not the case, the shortest element would have to be taken for the calculation of the step. With respect to the propagation speed, it depends on the mechanical characteristics of the beam section. However, the authors have deduced the critical integration step by trial and error so that, when the integration step is above the critical time, the algorithm degenerates. Otherwise, the algorithm gives reasonable results. Therefore, different increments of $t$ have tried to determine the critical value below when the algorithm is stable, and the results are feasible.

Thus, for this proposal, the speed and acceleration in the coordinates are initially approximated for a moment of time $t_n$, and an interval $\Delta t$, in accordance with:

$$\begin{aligned} \dot{q}_n &= \tfrac{1}{2\Delta t}(q_{n+1} - q_{n-1}) \\ \ddot{q}_n &= \tfrac{1}{\Delta t^2}(q_{n+1} - 2q_n + q_{n-1}) \end{aligned} \tag{16}$$

Substituting these equations into Equation (1) yields the following expression:

$$\left( \frac{1}{\Delta t^2} M + \frac{1}{2\Delta t} C_n \right) q_{n+1} = R_n - K_n q_n - \varnothing_n^t \lambda_n + \frac{1}{\Delta t^2} M(2q_n - q_{n-1}) + \frac{1}{2\Delta t} C_n q_{n-1} \quad (17)$$

Equation (17) allows the coordinates $q_{n+1}$ at time $t_{n+1}$ to be obtained from the elements of the equation and the coordinates at the previous instant $t_n$. However, for this calculation, it is necessary to solve a system of linear equations because, although the mass matrix $M$ is diagonal, the same does not occur with the damping matrix $C_n$. After all, we assumed a Rayleigh damping model for the catenary according to Equation (4). Although it is possible to assume a diagonal matrix for $C_n$, we preferred to use the Rayleigh model because we think it is a more realistic approach. The same problem exists for the pantograph–damping matrix.

One of the main advantages of the explicit integration methods is that the integration variables can be obtained directly without having to solve a system of equations, resulting in a more computationally efficient integration. However, for this purpose, it is necessary to slightly modify the dynamic equation by correcting the velocity vector in the coordinates by half a step, which results in:

$$\begin{pmatrix} M & 0 \\ 0 & 0 \end{pmatrix} \begin{pmatrix} \ddot{q}_n \\ \ddot{\lambda}_n \end{pmatrix} + \begin{pmatrix} C_n & 0 \\ 0 & 0 \end{pmatrix} \begin{pmatrix} \dot{q}_{n-\frac{1}{2}} \\ \dot{\lambda}_{n-\frac{1}{2}} \end{pmatrix} + \begin{pmatrix} K_n & \varnothing_n^t \\ \varnothing_n & 0 \end{pmatrix} \begin{pmatrix} q_n \\ \lambda_n \end{pmatrix} = \begin{pmatrix} R_n \\ 0 \end{pmatrix} \quad (18)$$

Equation (18) involves the calculation of the inverse of a rank-deficient mass matrix. However, the algorithm actually works by firstly calculating the position of the nodes, for which the values of the mass matrix are non-zero following the method of central differences. This method appears in [23]. Thus, to calculate the inverse of a diagonal matrix of non-zero elements, it is only necessary to calculate the inverse of the particular element $m_i$, that is, $1/m_i$.

Once these positions are calculated, the contact forces are calculated for which the mass matrix is no longer necessary following Hooke's law where the nodes, which are the ends of the contact spring, and the stiffness of the spring are known.

In Equation (18), these terms appear together although they can be calculated, as mentioned above, in a decoupled manner.

According to reference [23], this approach introduces a small error that can be ignored in structural systems with low damping, as seen in our case. Next, the velocity and acceleration in the coordinates are approximated by the following equations:

$$\dot{q}_{n-\frac{1}{2}} = \frac{1}{2\Delta t}(q_n - q_{n-1})$$
$$\ddot{q}_n = \frac{1}{2\Delta t}\left( \dot{q}_{n+\frac{1}{2}} - \dot{q}_{n-\frac{1}{2}} \right) = \frac{1}{\Delta t^2}(q_{n+1} - 2q_n + q_{n-1}) \quad (19)$$

Substituting the previous equations into Equation (18) and solving for the vector $q_{n+1}$ yields:

$$\left( \frac{1}{\Delta t^2} \right) M q_{n+1} = R_n - K_n q_n - \varnothing_n^t \lambda_n + \frac{1}{\Delta t^2} M \left( q_n + \Delta t \dot{q}_{n-\frac{1}{2}} \right) - C_n \dot{q}_{n-\frac{1}{2}} \quad (20)$$

In this case, it is possible to calculate the vector $q_{n+1}$ directly without the need to solve a system of equations because the mass matrix is diagonal. However, some variables remain to be determined: the position of the contact elements $(y_3)_{n+1}$ and $(y_4)_{n+1}$, and contact forces $(\lambda_1)_{n+1}$ and $(\lambda_2)_{n+1}$. The mass of the contact elements is null and Equation (20) cannot be applied; however, the positions of the catenary nodes in the pantograph environment are known at $t_{n+1}$. Thus, when the contact elements are activated, we use Equation (10) particularized in $t_{n+1}$ for $(y_3)_{n+1}$ and a similar equation for $(y_4)_{n+1}$. When the contact element is deactivated, its position coincides with that of the head mass; in this case, we will use Equation (14) or (15).

For the contact forces in $t_{n+1}$, these forces are calculated according to the deformation in the corresponding contact spring, according to the expressions:

$$(\lambda_1)_{n+1} = k_{cont}\left[(y_3)_{n+1} + v_1 - (y_2)_{n+1}\right], \quad (\lambda_2)_{n+1} = k_{cont}\left[(y_4)_{n+1} + v_2 - (y_2)_{n+1}\right] \tag{21}$$

As can be seen in the previous equations, an additional correction term is added, given by the variables $v_1$ and $v_2$, to the theoretical deformation of the spring given by the difference in the positions of the contact element and the head mass. This term takes into account the effect of the slope in the sectioning span as a consequence of the previous configuration of the beam. The values of functions $v_1$ and $v_2$ are discussed in the following section.

Because the contact force is a compression force, each integration step in Equation (21) must be solved by checking that the forces are negative. In this case, the value of the force is correct and the corresponding stiffness of the contact spring $k_3$ or $k_4$ is the theoretical value $k_{cont}$ of the stiffness matrix in Equation (7), and this value is maintained for the next integration step. However, if a value of the force is positive, it means that the spring works in traction, and therefore a take-off, has taken place. In this case, the contact force becomes zero and the corresponding rigidity is annulled for the next step. This is equivalent to eliminating the contact catenary–pantograph. Based on the above, the algorithm for detecting take-off at instant $t_{n+1}$ is:

$$\begin{aligned}
&if \ (\lambda_1)_{n+1} \ \leq \ 0 \ \rightarrow (k_3)_{n+1} = k_{cont}; \ if \ (\lambda_1)_{n+1} \ > \ 0 \ \rightarrow (k_3)_{n+1} = 0, \ (\lambda_1)_{n+1} = 0 \\
&if \ (\lambda_2)_{n+1} \ \leq \ 0 \ \rightarrow \ (k_4)_{n+1} = k_{cont}; \ if \ (\lambda_2)_{n+1} \ > \ 0 \ \rightarrow \ (k_4)_{n+1} = 0, \ (\lambda_2)_{n+1} = 0
\end{aligned} \tag{22}$$

According to Equation (22), it takes into account the possibility of contact and take-off by considering the impact between the catenary and the pantograph. Thus, for catenary 1, if the contact force is negative or equal to 0 ($(\lambda_1)_{n+1} \leq 0$), then there is contact, and the stiffness matrix is configured according to this condition for the next step. If the contact force is positive, no contact is considered, the contact force is 0, and the stiffness matrix is set for the next step with that condition.

For the second catenary, it is the same procedure ($(\lambda_2)_{n+1} \leq 0$).

It is evident that, when the pantograph does not circulate through the common transition area, some contact elements are deactivated, and the corresponding force will be annulled. In addition, when circulating through the common transition area, the two contact elements are activated, and take-offs and couplings can be detected in all cases. To initialize the integration algorithm, it is necessary to provide a value for the coordinates and their derivatives, and for the contact forces in the zero instant. It is assumed that, at the initial instant, the velocities and accelerations in the coordinates are zero:

$$\dot{q}_0 = \dot{q}_{-\frac{1}{2}} = \ddot{q}_0 = 0 \tag{23}$$

Substituting these values into Equation (18) results in a linear system that can be resolved for $q_0$ and $\lambda_0$, as follows:

$$\begin{pmatrix} K_o & \varnothing_0^t \\ \varnothing_0 & 0 \end{pmatrix} \begin{pmatrix} q_0 \\ \lambda_0 \end{pmatrix} = \begin{pmatrix} R_0 \\ 0 \end{pmatrix} \tag{24}$$

Equations (19)–(22), which are based on the method of explicit integration of central differences, make it possible to obtain the system variables directly by simply treating the non-linearities caused by take-offs and the effect on the overlapping, thus adapting the different variable elements of the dynamic equation at each integration step.

### 2.8. Functions for Correcting the Contact Position Due to the Effect of Overlapping Span

Although the assembly of a rigid catenary canton is undertaken using a series of spans in which the beams are assumed to be straight, in the case of overlapping spans, it may be advisable to previously deform the beam by giving it a sloped shape, and thus obtain the smoothest possible transition between cantons. This makes it difficult to simulate because the finite element model used is based on the hypothesis of straight beams. To overcome

this disadvantage, it is assumed that the rigidity of the beam is not altered by the previous deformation. Thus, the integration of the differential Equation (18) is carried out as if the catenary is configured as a perfectly straight beam, both in the normal spans and in the section span. Then, to take into account the previous deformation of the beam and its influence on the contact force, the vertical position of the contact element calculated from Equation (10) is supplemented by the correction functions $v_1$ and $v_2$. These functions take into account the variation of the position of these elements to then calculate the contact forces according to Equations (21) and (22). These functions depend on the geometry of the deformation and their effect is only taken into account when the pantograph is circulating through the section slopes. In the simulation proposed in the work, a linear slope is also assumed due to its simplicity; however, this methodology can also be adapted to any beam configuration with slight modifications.

In the case of a linear slope, the variable $x$ defines the horizontal displacement of the pantograph from the beginning of the route, $x_1$ is the position where the ascending slope begins in the canton of outgoing catenary 1, and $x_2$ is the starting position of the descending slope in the canton of incoming catenary 2, with $x_1 > x_2$, as shown in Figure 7. Let $L_1$ be the length of catenary slope 1, $L_2$ be the length of catenary slope 2, and $h_1$ and $h_2$ be the respective heights at the end of each slope. This is compiled for $v_1$ as:

$$
\begin{aligned}
u_1 &= x - x_1 \\
x < x_1 \cup x > x_1 + L_1, \quad v_1 &= 0 \\
x \geq x_1 \cap x \leq x_1 + L_1, \quad v_1 &= \tfrac{h_1}{L_1} u_1
\end{aligned}
\tag{25}
$$

and for $v_2$ as:

$$
\begin{aligned}
u_2 &= x - x_2 \\
x < x_2 \cup x > x_2 + L_2, \quad v_2 &= 0 \\
x \geq x_2 \cap x \leq x_2 + L_2, \quad v_2 &= h_2 - \tfrac{h_2}{L_2} u_2
\end{aligned}
\tag{26}
$$

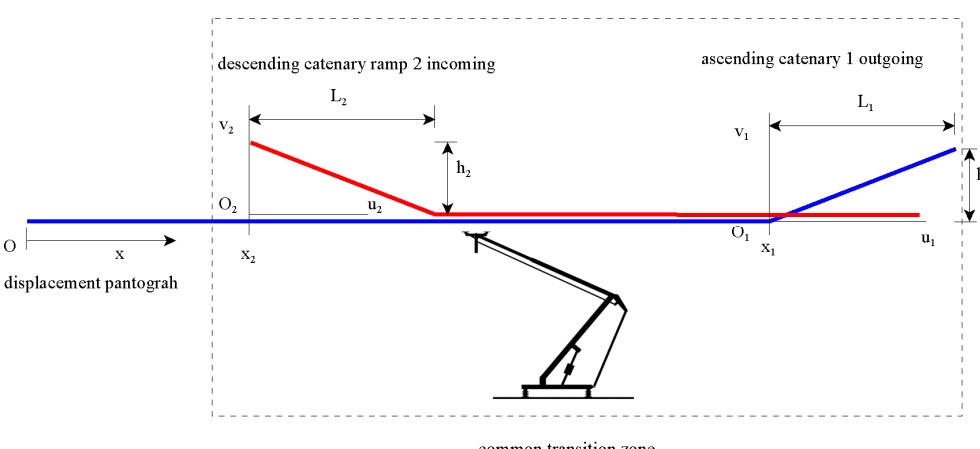

**Figure 7.** Position correction functions of the contact element for linear slope.

### 2.9. Algorithm for the Integration of Dynamic Equation

According to the description in the previous sections, the algorithm for the integration of the dynamic equation in the modified form of Equation (18) is as follows:

1.  Input data for catenary and pantograph. Regarding the catenary: the number of spans of the cantons, lengths of spans, moments of inertia of the elements of the Furrer beam section (body of the beam and the contact wire), characteristics of the materials, the geometry of the sections, characteristics of the flanges, etc. The characteristics of the equivalent section of the beam are then determined according to Equations (5) and (6). For pantographs, the following are entered: number and type of pantographs, masses, spring stiffnesses, dampers, etc.

2. The setting of the different terms of the dynamic equation at the initial instant: the stiffness matrix, damping matrix, mass matrix, and external load vector, according to Equations (2)–(4) and (7). The matrix of restriction conditions at the initial position of the pantograph(s) is also established according to Equations (8)–(15).

3. The initial value of the integration variables is set to $t_0 = 0$: coordinates of the nodes $q_0$ and their derivatives, and the initial contact force $\lambda_0$ according to Equations (23) and (24).

4. The integration cycle begins at the time of the differential Equation (18) in $t_{n+1}$. The cycle varies for different time values with $n = 0,1,2 \ldots$ , adding in each cycle the value of $\Delta t$ to $t_n$:

   1. Calculation of the coordinates corresponding to the positions of the catenary nodes and the pantograph masses $q_{n+1}$, according to Equation (20).
   2. Calculation of the position of the contact elements of the pantograph(s): $(y_3)_{n+1}$ and $(y_4)_{n+1}$ according to Equation (10). If the contact element is deactivated, its position coincides with that of the pantograph head mass; in this case, Equation (14) or (15) shall be used.
   3. Calculation of the pantograph–catenary contact forces $(\lambda_1)_{n+1}$ and $(\lambda_2)_{n+1}$, according to Equations (21) and (22).
   4. Setting the restriction conditions matrix $\phi_{n+1}$, using Equations (8)–(15), according to the advance of the pantograph(s) along the line.
   5. For the calculation of the forces, the positions of the contact elements, and the restriction conditions, the three cases of activation of the contact element are accounted for, according to the position of the pantograph along the line, as explained in Section 2.6.
   6. The remainder of the terms of the dynamic Equation (18) is also set in $t_{n+1}$: $C_{n+1}$, $K_{n+1}$, and $R_{n+1}$, and, in particular, the pantograph stiffness matrix must be modified if there is a take-off or a coupling according to Equation (22).
   7. Calculation of the velocity vector in the coordinates for the next integration cycle in $t_{n+1/2}$ by the mid-pass correction of the velocity vector, using Equation (19).

## 3. Results

As a result of the previous study, a software tool called RICATI was developed. In this section, the results obtained from this software and the simulation are discussed.

### 3.1. Computational Aspects

To facilitate the use of the algorithms, the RICATI application was created, which incorporates the algorithms with a user-friendly interface for the introduction of values from a database. The tool also presents graphic results that allow conclusions to be obtained without having to directly interpret a table with hundreds of thousands of numerical data points [26].

In order to achieve an efficient software tool, the enhancements carried out in this work are related to the way of storing the matrices involved in the resolution of the dynamic equation, as well as the use of suitable methods for solving the associated static problem.

In particular, the sparsity and symmetry of the stiffness matrix are exploited, improving the efficiency of the implementation, dramatically reducing the memory storage requirements, and making use of iterative methods based on Krylov projection methods. The execution time spent for solving the static problem is also dramatically reduced. For this purpose, the SPARSKIT library by Yousef Saad is used [27].

Therefore, a high-performance implementation has to take into account the features of current architectures, for example, cache memory. These features are particularly important when rebuilding the traditional algorithms to a block-oriented implementation. Block-oriented algorithms reduce drastically the data flow between main memory and secondary memory enhancing the performance of the final implementation. These good features are obtained by using standard libraries in the implementations. In particular, for this paper,

several computational kernel from BLAS library [28,29], especially BLAS 3, are identified and used.

RICATI executes the algorithms based on the data of two cantons with the number of spans indicated for each canton. As a result, an interactive graphic interface is provided that allows the behavior of the catenary to be observed before the passage of the pantograph.

RICATI is structured into four basic components, as shown in Figure 8:

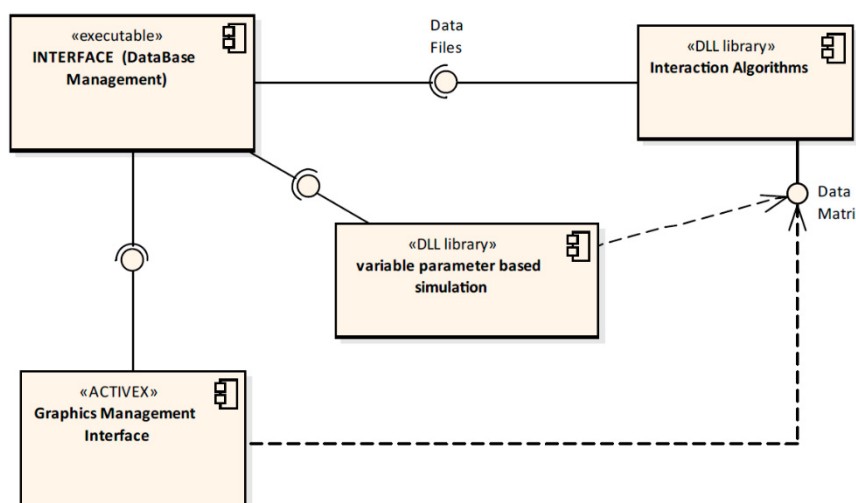

**Figure 8.** Component diagram of the RICATI application.

1.  Interface: The interface allows all of the data associated with the assembly to be entered. The set of parameters is stored in a database so that they can be retrieved, modified, or copied to obtain different calculations or perform simulations. The interface is structured in blocks according to the kind of data that they contain:

    a.  General parameters: Generic data values are introduced regarding integration, speed, and data common to the whole catenary.
    b.  Values for the catenary damping frequency: The critical damping percentage is established for two reference frequencies.
    c.  Section data: Geometry of the overlapping span.
    d.  Cross-section properties: Mechanical properties of the profile, wire, and flanges.
    e.  Pantographs: Data of the pantograph to be used in the interaction. Up to four pantographs can be used. The pantographs are coded in such a way that they can be used in different calculations without the need to re-introduce all of their characteristics.
    f.  Canton spans: Number of spans in each canton and for each span, length and height of the support above the nominal value, and distance from the flange to the left support.

2.  Interaction Algorithms: The dynamic library containing the algorithms for calculating the pantograph–catenary interaction. As a result, among other values, a table is generated indicating at each instant, according to the defined integration time step, and for each pantograph, the vertical position of two catenary points preset in the calculation, the height of the contact point, and the effort in each canton and plate.

3.  Graph management: As a result, an interactive graphical interface is displayed in which different graphs can be obtained by independently selecting the abscissa (distance, time) and ordinate (efforts, elevation) axes, and allowing different graphs to overlap, in addition to the magnification of the area of the graphs marked by the user, as shown in Figure 9.

4.  Simulation: This allows for repeated execution of the calculation by modifying, at an established interval and increment, parameters such as the length of the overlapping span, the height of the slope, and the distance from the beginning of the slope to the

end of the slope. The application repeatedly executes the algorithm by varying these data, thus allowing for a table with the maximum, minimum, deviation, number of take-offs, total distance of take-offs, and total time of take-offs in each independent canton or considering the sum of the efforts in the common zone of the two cantons.

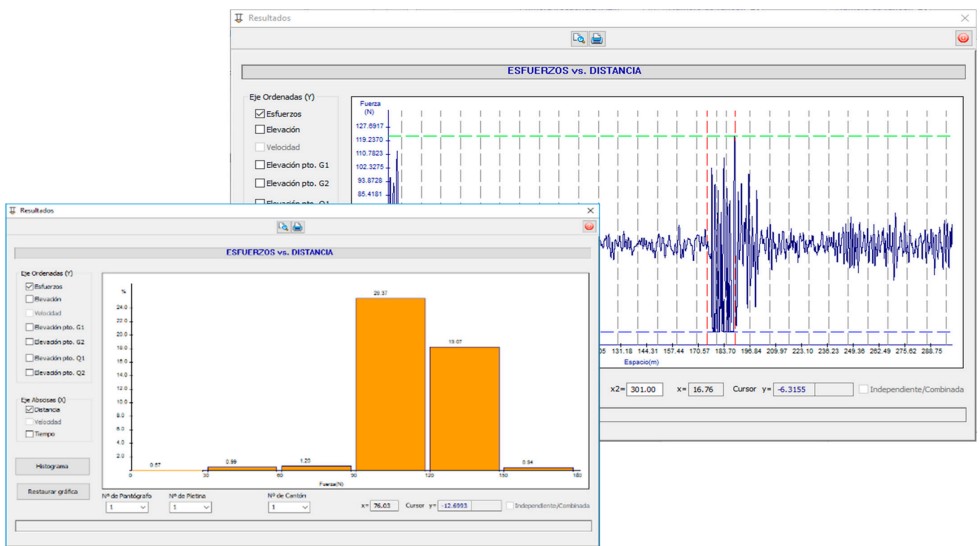

**Figure 9.** Graphical results of the RICATI application.

### 3.2. Simulation Results

The objective is to check the influence of the geometric characteristics of the slopes in the common area of the catenary before the passage of the pantograph. For the execution of the simulations, a set of two cantons with the following characteristics is defined as the basis for the calculations:

- Two cantons of 20 spans of 10 m length in the normal spans.
- Height above the track: 5 m.
- Maximum common area length: 12 m.
- Aluminum profiles and flanges corresponding to a Furrer beam per the values in Section 4.
- A pantograph of a simple plate and two stages defined according to EN50318 whose data are in Table 1 of Section 2.4.
- As in standard EN50318 for the flexible catenary, no damping is assumed for the catenary.
- The interface for the remainder of the values defining the catenary.

Using the RICATI application, we carried out different simulations in which the behavior of the catenary at the passage of the pantograph is evaluated according to values such as the mean, deviation of efforts, and take-offs in the common area.

A good result is considered to include a minimum deviation, no take-offs, and an average of the forces close to the pushing force of the pantograph (approx. 120 N).

To check the most suitable configuration, four simulations are carried out:

- An overlapping span of 1 m, slope length 1 m, and variation in the height of the slope at its end.
- An overlapping span of 1 m, variation in slope length in the range of 0.5 to 1 m with 0.05 m increments, and 0.1 m slope elevation.
- Variation of the length of the overlapping span in the range of 0.5 to 5 m and height 0 of the slope. Normal spans are 10 m long.
- Variation of the length of the overlapping span in the interval from 0.5 to 5 m and slope height 0. Normal spans are 5 m long.

### 3.2.1. Simulation 1

We consider an overlapping span of 1 m in both cantons beginning the slope of the slope from the beginning of the span. In each calculation, the height of the end of the slope varies from 0 m (fully horizontal) to 0.2 m with an increase of 0.005 m.

The analysis of the results is carried out in the common area and, as a result, we obtain the following graphs that represent the average values of the pantograph effort on the contact wire in Newtons, the standard deviation, and the average distance of take-offs. In this case, there are no take-offs in the common area of the catenary, so this graph is omitted.

The abscissa axis of the graphs represents the height of the slope at its end in meters, and the ordinate axis represents the effort (mean, maximum, and deviation) in Newtons, as obtained in the integration calculated with the given elevation.

By analyzing the graphs (Figure 10), it is observed that the deviation of the efforts in the common zone is greater because we increase the slope. Analogously, the average effort and the maximum values increase with a behavior similar to the variation of the deviation.

The effort of the pantograph passing through the common area has a better behavior in the case of zero elevation from the horizontal slope and, therefore, with a 0 slope before its installation. This may seem counterintuitive when considering that, due to the weight of the beam, it should have a certain negative slope and therefore be an obstacle for the passage of the pantograph. However, in the static configuration of the catenary assembly, the beam has a certain positive elevation due to the weight of the beam in the adjoining span, which would justify the results of this simulation.

We then observe that a rise added to the beam on the slope could eventually be counterproductive. However, this raised the question about the influence of the length of the sectioning span on these results. Thus, we proposed the second simulation.

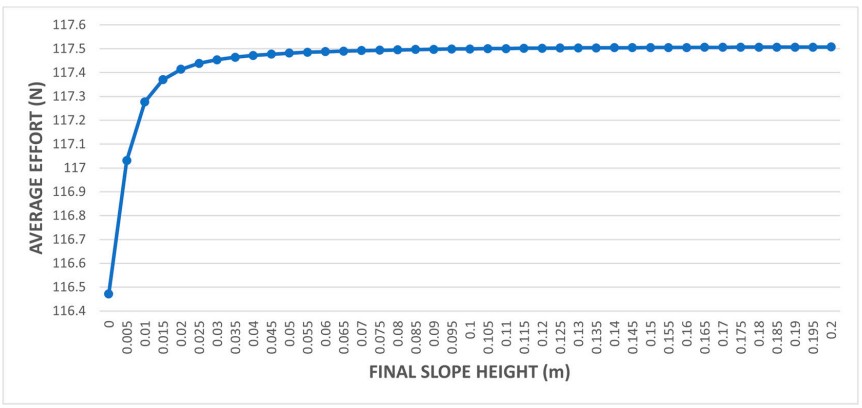

(**a**) Average value of pantograph effort.

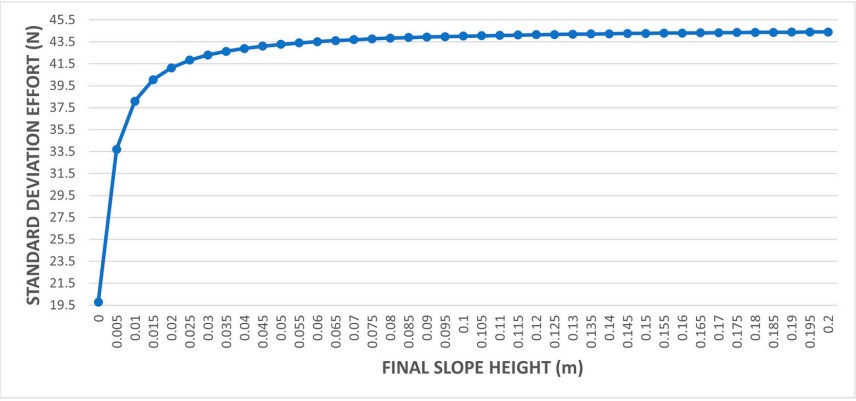

(**b**) Standard deviation of pantograph effort.

**Figure 10.** *Cont.*

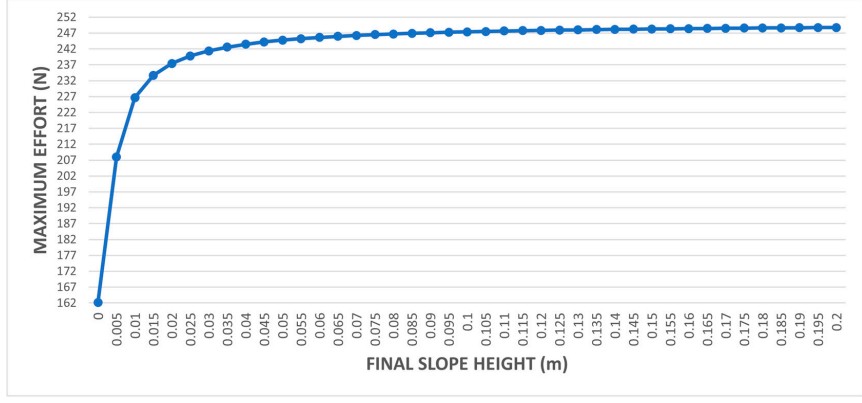

(**c**) Maximum value of pantograph effort.

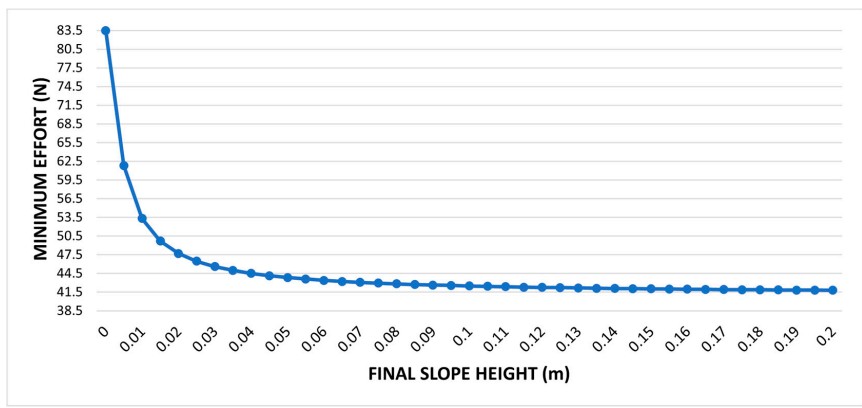

(**d**) Minimum value of pantograph effort.

**Figure 10.** Graphs of simulation 1: an overlapping span of 1 m, slope length 1 m, and variation in the height of the slope at its end. (**a**) Average value of pantograph effort, (**b**) standard deviation of pantograph effort, (**c**) maximum value of pantograph effort, and (**d**) minimum value of pantograph effort.

3.2.2. Simulation 2

We consider a sectioning span of 1 m in both cantons, while maintaining the elevation of the sloping end at 0.1 m. In each calculation, the length of the slope length is varied from 0.5 to 1 m with an increase of 0.05 m.

The analysis of the results is carried out in the common area and, as a result, we obtain the graphs shown in Figure 11, which represent the average, maximum, and minimum values of the pantograph effort on the contact wire in Newtons, the standard deviation, and the average distance of take-offs. In this case, there are no take-offs in the common area of the catenary, so this graph is omitted.

The abscissa axis of the graphs represents the length of the slope in meters and the ordinate axis represents the effort (mean, maximum, minimum, and deviation) in Newtons obtained in the integration calculated with the given elevation.

Analysis of the graphs (Figure 11) shows that the deviation of the stresses in the common area is adequate in cases in which the length of the slope is less than or equal to 0.45 m. The deviation of the stresses in the common area is adequate in cases in which the length of the slope is less than or equal to 0.45 m. From this length, we see how the maximum and the deviation increase as the length of the slope increases until it coincides with the whole section span. Similarly, the minimum stresses decrease as the maximum increases from 0.45 m in slope length, confirming the increase in deviation.

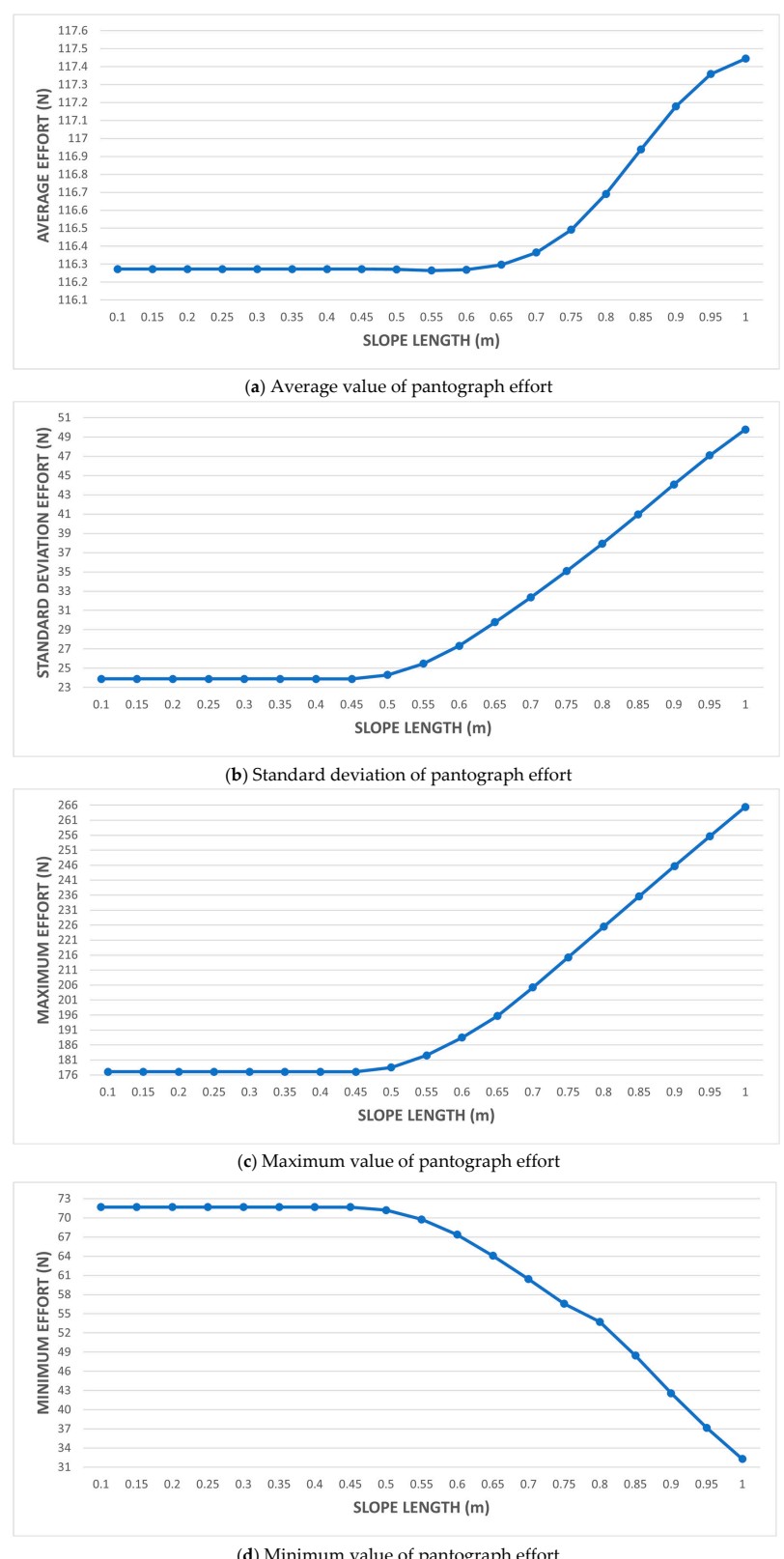

(**a**) Average value of pantograph effort

(**b**) Standard deviation of pantograph effort

(**c**) Maximum value of pantograph effort

(**d**) Minimum value of pantograph effort

**Figure 11.** Graphs of simulation 2. Overlapping span of 1 m variation in slope length in the range of 0.5 to 1 m with 0.05 m increments and 0.1 m slope elevation. (**a**) Average value of pantograph effort, (**b**) standard deviation of pantograph effort, (**c**) maximum value of pantograph effort, and (**d**) minimum value of pantograph effort.

The effort at the passage of the pantograph through the common area has a better behavior in the case of small lengths of the slope (considering an elevation of 0.1 m from the end). This result is congruent with simulation 1 because the case of 0 m of the slope is equivalent to a horizontal overlapping span in which only the elevation obtained in the initial static configuration is applied. We then observe that the length of the overlapping span can significantly influence the results obtained for a small elevation of the beam and that it could be sufficient with the elevation obtained in the initial static configuration. In this case, the length of the overlapping span profile influences the elevation of the beam and, therefore, the results obtained.

To test this hypothesis, the following two simulations start from the 0 m elevation of the slope and the variation of the overlapping span length. In simulation 3, we consider a configuration equal to that used previously (i.e., normal spans of 10 m), and in simulation 4 we change the length of the normal spans to 5 m to check the influence of the length of the normal spans on the result.

### 3.2.3. Simulation 3

We consider an overlapping span of 0.5 to 5 m in both cantons with an increase of 0.1 m in each calculation. In all cases, a slope elevation of 0 m is considered, i.e., we use a horizontal beam without slope for overlapping span to obtain the optimum length of the beam to establish a suitable elevation. We continue with the same configuration of normal spans of 10 m.

As in the previous cases, the analysis of the data obtained by the calculations is carried out in the common area, and, as a result, we obtain the graphs shown in Figure 12, which represent the average and maximum values of the pantograph stress on the contact wires in Newtons, the standard deviation, and the total distance of take-offs. In this case, and as expected, take-offs may occur in the common area of the catenary; thus, we show the graph indicating the total length of take-offs produced in this area.

The abscissa axis of the graphs represents the length of the overlapping span in meters and the ordinate axis represents the effort (mean, maximum, minimum, and deviation) in Newtons obtained in the integration calculated with the given elevation.

Analysis of the graphs (Figure 12) shows that the deviation of the stresses in the common zone is adequate in cases in which the span length is below 3.5–3.7 m, with slightly worse behavior at values below 1.5 m. Above 3.7 m, the behavior worsens significantly, and take-offs appear over longer distances. This behavior is explained by the absence of the section elevation effect compensated by the "excessive" length of the overlapping span, which causes it to lose the elevation and fall below the nominal height on the track. Thus, the pantograph "collides" with this beam with stresses of more than 2000 N.

In the configuration analyzed, the optimum distance is around 3.5 m. From this length onwards, it is necessary to elevate the slope to achieve optimum values in the common area of the catenary. However, if we consider the results separated by the canton represented in Figure 13, we observe that, from 3.1 m, take-offs occur in the wire of canton 1 and, from 3.4 m, in the wire of canton 2. To ensure that the optimum distance is below these values, it is necessary to maintain contact of the plate with the two wires simultaneously in the common area.

It is expected that the optimum distance of the overlapping span length is influenced by the length of the adjacent normal spans. We can check this situation by repeating this simulation with normal spans of 5 m instead of 10 m, as shown in the following simulation.

### 3.2.4. Simulation 4

We consider a sectioning span of 0.5 to 5 m in both cantons with an increase of 0.1 m in each calculation. In all cases, a slope elevation of 0 m is considered, i.e., we use a horizontal slope in the sectioning to obtain the optimum length of the beam and, thus, to ensure a suitable configuration. In this case, the normal spans of both cantons are considered to have a length of 5 m to check the differences with the previous simulation.

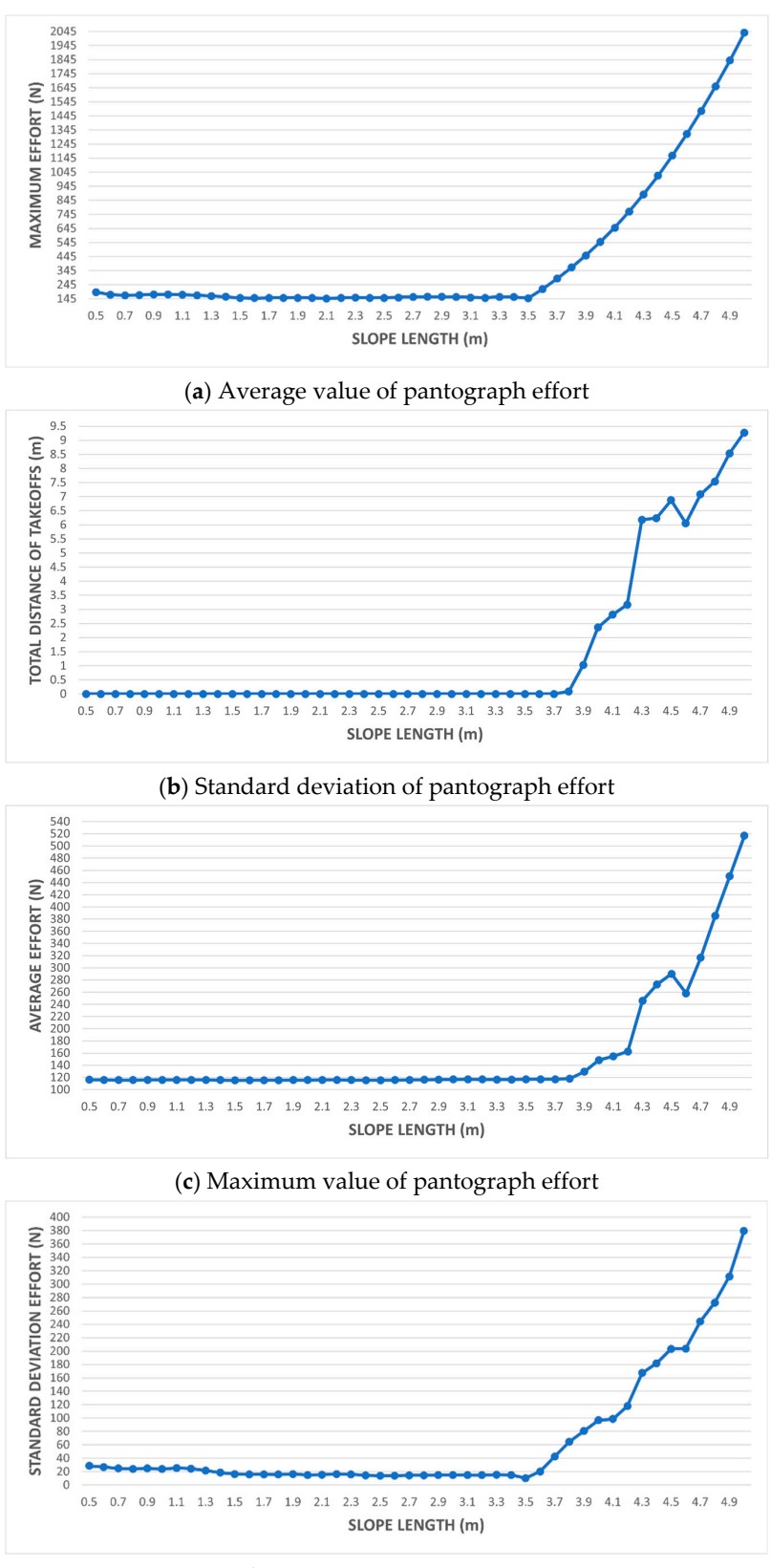

(**a**) Average value of pantograph effort

(**b**) Standard deviation of pantograph effort

(**c**) Maximum value of pantograph effort

(**d**) Total distance of take-offs

**Figure 12.** Graphs of simulation 3. Variation of the length of the overlapping span in the range of 0.5 to 5 m and height 0 of the slope. Normal spans are 10 m long. (**a**) Average value of pantograph effort, (**b**) standard deviation of pantograph effort, (**c**) maximum value of pantograph effort, and (**d**) total distances of take-offs.

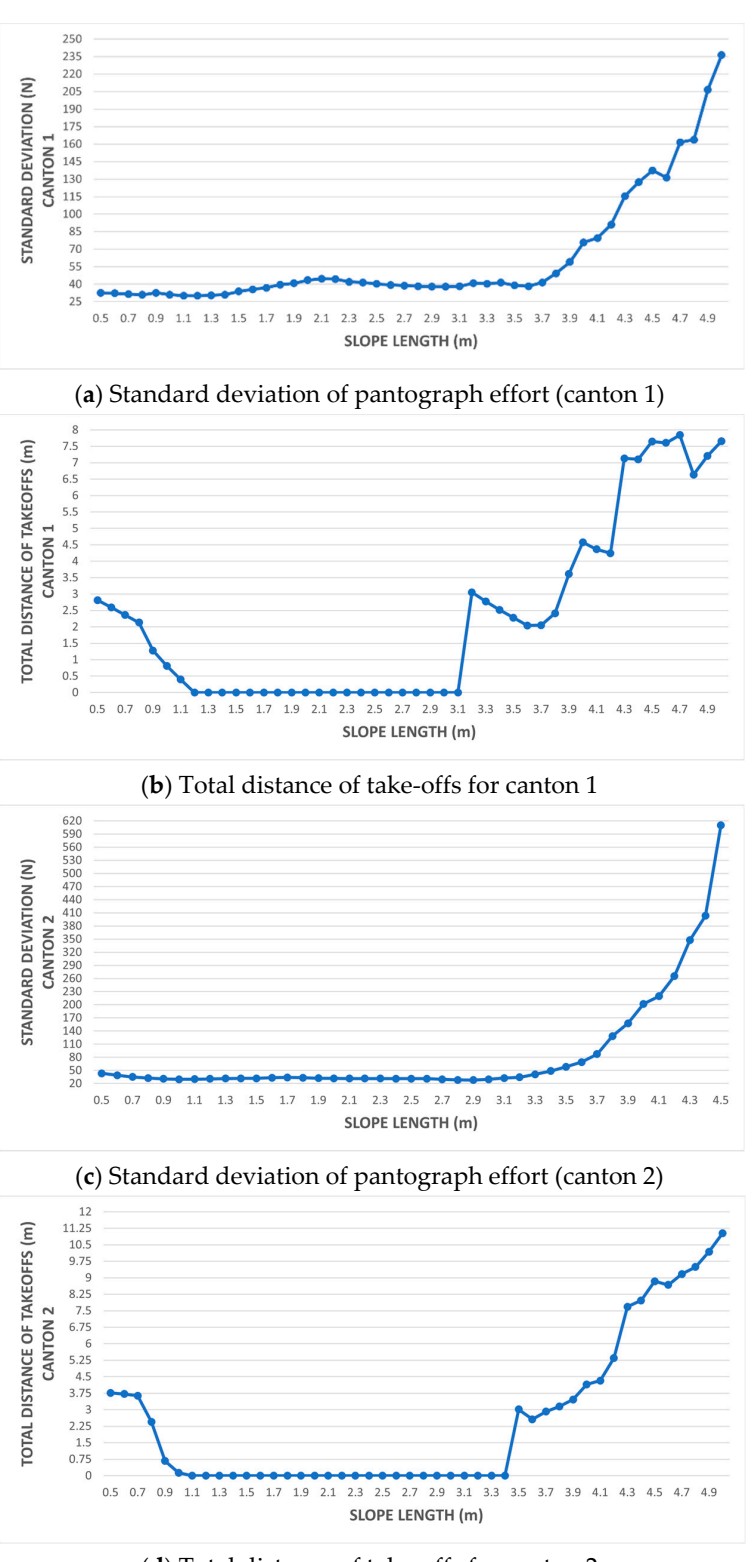

(**a**) Standard deviation of pantograph effort (canton 1)

(**b**) Total distance of take-offs for canton 1

(**c**) Standard deviation of pantograph effort (canton 2)

(**d**) Total distance of take-offs for canton 2

**Figure 13.** Graphs of simulation 3 per cantons. Variation of the length of the overlapping span in the range of 0.5 to 5 m and height 0 of the slope. Normal spans are 10 m long. (**a**) Standard deviation of pantograph effort (canton 1), (**b**) total distances of take-offs for canton 1, (**c**) standard deviation of pantograph effort (canton 2), and (**d**) total distances of take-offs for canton 2.

As in the previous cases, the analysis of the data obtained by the calculations is carried out in the common area and, as a result, we obtain the graphs shown in Figure 14, which represent the average, maximum, and minimum values of the pantograph effort on the

contact wire in Newtons, the standard deviation, and the average distance of take-offs. In this case, and as expected, take-offs may occur in the common area of the catenary; thus, we show the graph indicating the total length of take-offs produced in this area.

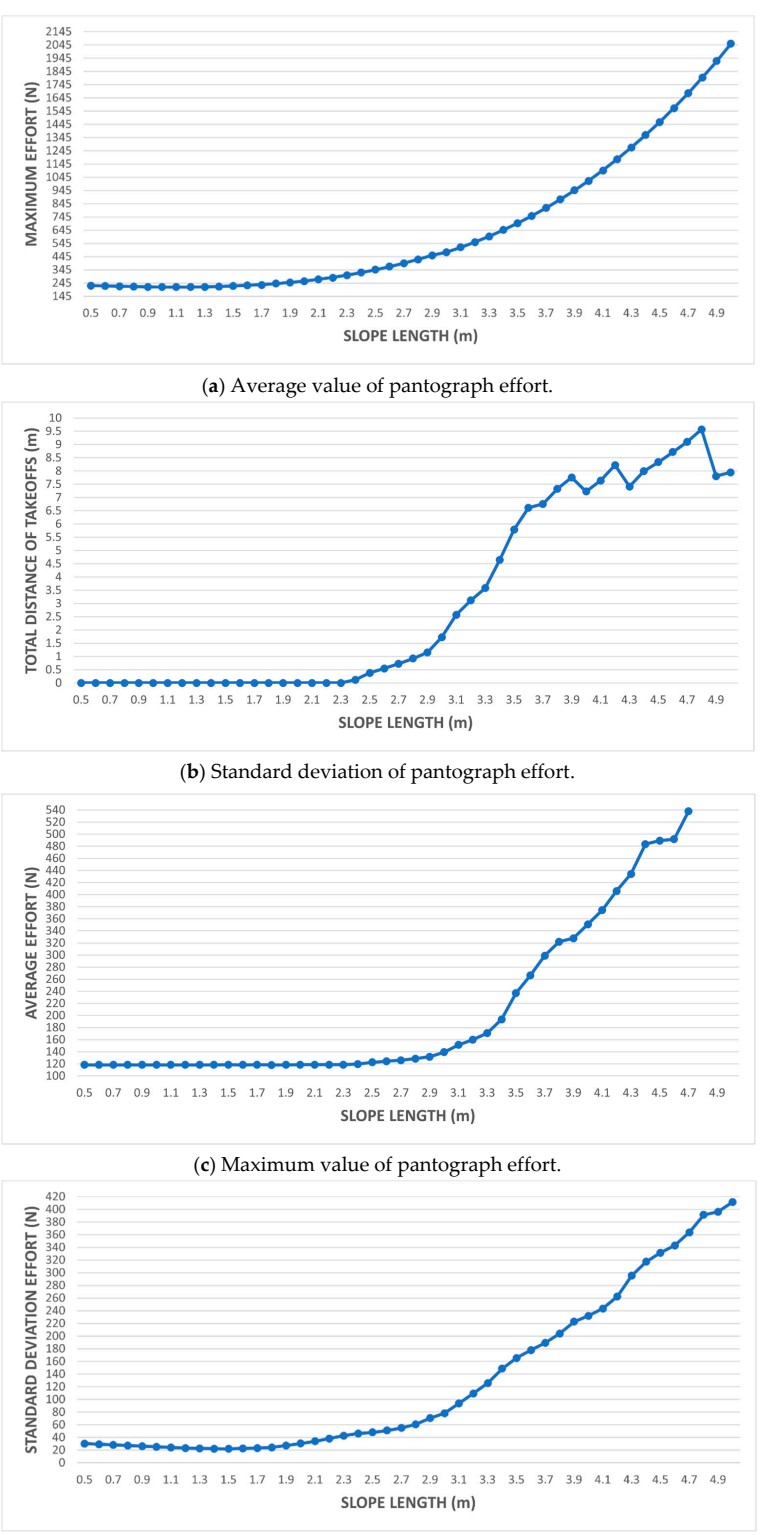

(**a**) Average value of pantograph effort.

(**b**) Standard deviation of pantograph effort.

(**c**) Maximum value of pantograph effort.

(**d**) Total distance of take-offs.

**Figure 14.** Graphs of simulation 4. Variation of the length of the overlapping span in the interval from 0.5 to 5 m and slope height 0. Normal spans length of 5 m. (**a**) Average value of pantograph effort, (**b**) standard deviation of pantograph effort, (**c**) maximum value of pantograph effort, and (**d**) total distances of take-offs.

The abscissa axis of the graphs represents the length of the overlapping span in meters and the ordinate axis represents the effort (mean, maximum, minimum, and deviation) in Newtons obtained in the integration calculated with the given elevation.

An analysis of the graphs (Figure 14) shows that the deviation of the stresses is optimal for the span length of approximately 1.4 m or less, with slightly worse behavior at values below 1 m. From approximately 2 m, the behavior begins to worsen significantly and take-offs appear over greater distances.

The behavior is similar to that of simulation 3, but because the normal spans are shorter, the effect of lifting the beam in the section is dissipated before the critical point at 1.4 m, from which the lifting of the beam is compensated by starting to fall below the nominal height, which causes problems with the passage of the pantograph.

## 4. Discussion

The inclusion of the slope in an overlapping span may be necessary from a critical length onwards to achieve optimum values of force in the common area. However, in general, for the assumed speed, it is not necessary to introduce a slope height to obtain good results. This is because, in the initial static configuration, the section already presents a certain inclination as a consequence of the action of gravity. However, the length of the section should not exceed a certain value from which the results of the contact force significantly fluctuate.

As the authors state at the beginning of this paper, the main contribution of this paper consists in a model for the pantograph–rigid catenary interaction considering an overlapping span. This model is a particular model for this kind of catenary and not an elastic catenary where the stiffness is increased.

In order to build this model, the authors start from well-known methods that appear in Finite Element Method literature, such as [22,23], but particularized for the elements of a rigid catenary and its underlying problems. In fact, according to the literature, the model uses an explicit method as an integration method.

The RICATI tool allows the study of dynamic behavior in which new profile designs are considered and compared with existing designs. Therefore, RICATI allows multiple simulations to determine the optimal catenary structure for any configuration. For example, with the same characteristics as the previous simulations, it is possible to vary the speed of the unit to observe the behavior at different speeds. We observed that, at speeds of 80 and 120 km/h, the behavior of the examples is similar. However, at 200 km/h, Figure 15 shows variations in the behavior and indicates two optimal points in the length of the sectioning around 1.2 and 3.9 m.

This example shows the usefulness of an application, such as RICATI, to check the behavior of the initially established configuration for a catenary. The tool also allows solutions to be obtained for the problems encountered when simulating the passage of the pantograph (or pantographs), not only for the section, but also for the entire catenary.

The proposed methodology provides a tool to simulate the pantograph–catenary behavior in a rigid catenary considering the transition between two cantons of the catenary with overlapping spans. To advance the presented method, the following lines of work are proposed:

- Study of the behavior for different configurations in the geometry of the sectioning span.
- Development of the model in three dimensions.
- Study of the behavior and simulation in the transition between elastic and rigid catenary.
- Optimization of the algorithm, taking into account the data structures underlying the problem, in addition to the parallelization of the calculations, to obtain results in significantly shorter times.
- Development of RICATI following the paradigm of Software-as-a-Service. This model of the application in the cloud would allow access by more companies, from any point at which there is connectivity, and at a lower cost. In addition, this solution would

allow different tests to be carried out simultaneously by requesting resources from the provider of the cloud service, and significantly more quickly, to find better solutions.

In addition, as an important future work, the authors will address the problem of the irregularities in the beam following similar ideas of those described in [30–32].

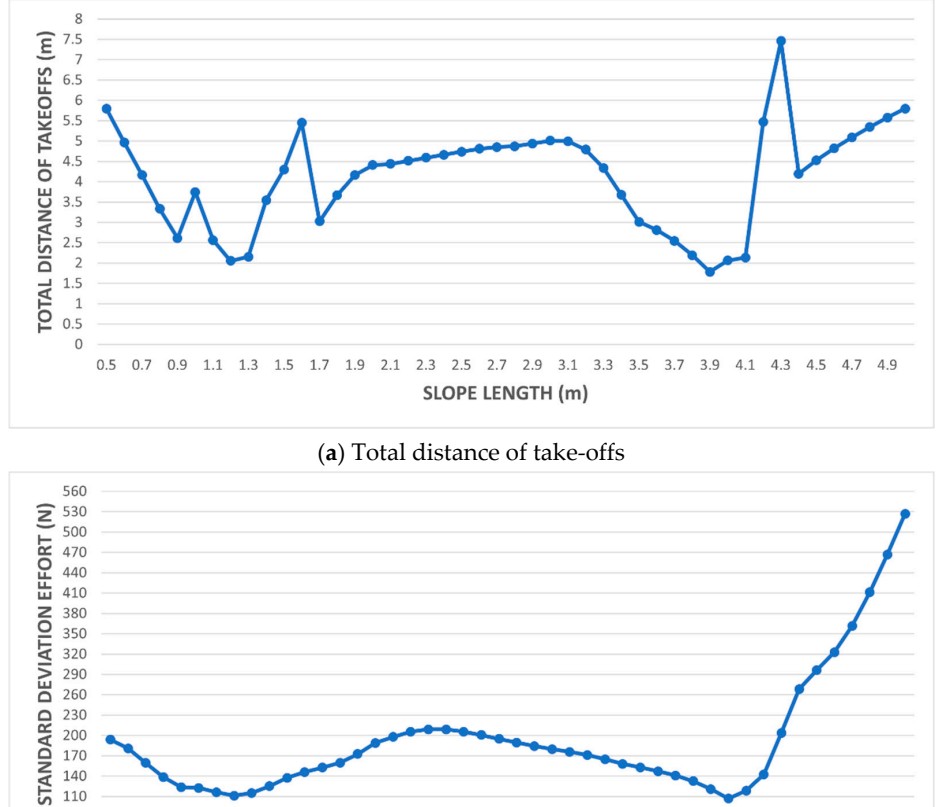

(**a**) Total distance of take-offs

(**b**) Standard deviation of pantograph effort

**Figure 15.** Graphs at total distance take-offs (**a**) and standard deviation (**b**) when the train is circulating at a speed of 200 km/h.

**Author Contributions:** J.B. contributed to the conceptualization, methodology, investigation, and writing—original draft. F.C. contributed to software, methodology, writing—original draft, and supervision. T.R. contributed to the software, methodology, validation, investigation, and writing—original draft. P.T.: software, validation, and investigation. E.A. contributed to the software, methodology, investigation, and writing—original draft. All authors have read and agreed to the published version of the manuscript.

**Funding:** This research received no external funding.

**Conflicts of Interest:** The authors declare no conflict of interest.

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
