# Peer review of "A Dynamic Model for the Study and Simulation of the Pantograph–Rigid Catenary Interaction with an Overlapping Span"

_applsci, doi:10.3390/app11167445_

Round 1
Reviewer 1 Report
Summary
This study focuses on the relatively short and rigid catenary. On the catenary, the sudden variation of the contact force occurs. To simulate the catenary and pantograph interactive dynamics, a dynamic model is presented in this manuscript. The novelty of this work is the application of well-known methods to study the dynamic interaction.
General Comment
The reviewer cannot recommend this article for publication because of the lack of the novelty from the viewpoint of the simulation theory. In the introduction, the authors stated that “the main novelty lies in the application of well-known methods.” The application of the well-known (i.e., existing) methods cannot be considered as the novelty in the reviewer’s opinion. Also, the development of software cannot be considered as the novelty. It is necessary to propose a new simulation theory when the research is submitted to this class of journals.
The reviewer hopes the following comments will help the authors improve this work.
Comment 1
Line 41: few scientific studies have been conducted on this problem.
It is very important to state the reason why few scientific studies have been conducted to show the authors’ novelty. For example, "the sudden variation of the contact force easily causes a numerical divergence, and thus this problem cannot be numerically investigated so far."
Comment 2
Line 110: In this work, the rigid catenary was modeled as a flexible beam.
It is better to state the reason why the rigid catenary is modeled as a flexible beam. From the viewpoint of the computational cost, the rigid-body modeling is more suitable.
Comment 3
Line 136: These values were obtained experimentally by trial and error.
The experimental result or reference should be added to show how the damping coefficient is determined. In this manuscript, both of them are not presented.
Comment 4
Line 209: Notice that the mass matrix is rank deficient; that is, it is not possible to calculate its inverse. In this case, only the inverse of non-zero diagonal elements is calculated.
Most readers cannot understand this special calculation method dealing with a rank-deficient mass matrix. Appendix or reference are necessary to describe the method.
Comment 5
Line 297: Explicit methods are more suitable than implicit methods for dealing with non-linearities because they correctly take into account the variation of the stiffness matrix.
This sentence would cause a serious argument. Many nonlinear solvers have employed implicit methods (e.g., implicit Newmark beta). The authors must provide the evidence to support this sentence.
Comment 6
It is better to clearly state the advantage of the developed simulation method in abstract, introduction, and conclusion.
Author Response
"Please see the attachment."

Reviewer 2 Report
I still have reservations concerning the scientific content of the article.
Especially the integration method looks quite unprofessional; has it ever been discussed with some numerical expert? Explicit methods certainly lead to easily solvable equations, but often they suffer from instability.
The explanation of the numerical method is quite imprecise: Suddenly the authors switch to indices with half integers; the sentence "According to reference [7], this approach introduces a small error" should be made much more precise; I guess the authors talk of the local discretization error. What about giving some estimates of this error?
Also the description of the contact force calculation should be improved a lot. The same applies to the discussion of the straight and deformed beam elements in Sec. 2.8.
I want to repeat my advice, that screen shots are nt really appropriate for a scientific article; they would be adequate for a user manual.
Some minor remarks:
"we will Equations (14) or (15)."
The letters vi in equation (21) and in the following text look quite different.
Author Response
"Please see the attachment."

Reviewer 3 Report
The topic discussed in this paper is sure of interest for the scientific community and industry of pantograph-catenary interaction. But the reviewer still finds some issues which deserve to be properly tackled before the final publication.
1) My main concern regarding this paper is that the irregularity of the rigid catenary is not considered in the simulations. The big difference between the rigid catenary can the traditional soft catenary is that the former is more rigid. So, the behaviour of the rigid catenary is more like a track, instead of a soft wire. It is seen from some works on vehicle-track interaction [1-2] that the main disturbance to the vehicle is the irregularity on the rail surface. Even in the soft catenary system, the work in [3] also indicates that the irregularities of the catenary have a non-negligible effect on the contact force. For the rigid catenary, the deterioration of the current collection quality should be mainly caused by the irregularities. The authors should give a detailed summary of the relevant works (regarding the irregularities in track and soft catenaries) in the introduction and explain why this important factor is not included.
[1] W. Zhai, K. Wang, and C. Cai, "Fundamentals of vehicle-track coupled dynamics," Veh. Syst. Dyn., vol. 47, no. 11, pp. 1349-1376, 2009, doi: 10.1080/00423110802621561.
[2] Z. C. Zhang, J. H. Lin, Y. H. Zhang, W. P. Howson, and F. W. Williams, "Non-stationary random vibration analysis of three-dimensional train-bridge systems," Veh. Syst. Dyn., vol. 48, no. 4, pp. 457-480, 2010, doi: 10.1080/00423110902866926.
[3] Y. Song, Z. Liu, A. Rønnquist, P. Nåvik, and Z. Liu, "Contact Wire Irregularity Stochastics and Effect on High-Speed Railway Pantograph-Catenary Interactions," IEEE Trans. Instrum. Meas., vol. 69, no. 10, pp. 8196-8206, Oct. 2020, doi: 10.1109/TIM.2020.2987457.
2) Please comment on the reasonability to model the contact wire and the aluminium profile as a single body, as clearance may exist between two contact surfaces and may affect the dynamic behaviour.
3) The pantograph model has a contact stiffness of 50000 N/m as shown in Table 1. The reviewer understands that this value is from the old version of En 50318. But this standard is only for the validation of soft catenary, instead of a rigid one. In addition, the latest En 50318 (v. 2018) has already got rid of the assumption of 50000 N/m contact stiffness. Please comment on the reasonability of the selection of contact stiffness for rigid catenary-pantograph.
Author Response
"Please see the attachment."

Round 2
Reviewer 1 Report
General Comments
The reviewer appreciates the authors' effort to revise the manuscript. The manuscript still needs 3 major revisions and 4 minor revision. If these revisions are carefully treated, the reviewer's opinion will be changed from "Major Revision" to "Accept."
Major Revision 1
The methods used in this paper are well known. The methods used in this paper are not novel. However, the application of all of them together to solve the problem of pantograph-rigid catenary interaction is a novelty. Much of the work found in the literature has been developed for flexible catenary. But in this paper, existing methods and algorithms have been adapted in order to solve the case of pantograph-rigid catenary interaction. These results for rigid catenary, as well as the underlying application, constitute the novelty of this method.
Please copy and paste this kind of sentences in the introduction to clearly show what is novel and what is not novel.
Major Revision 2
Actually, indicating rigid catenary is a jargon language. In conventional (flexible) catenary, the model is based on a series of wires (support, contact wire and hangers) whereas in rigid catenary the model is a flexible beam but structurally it has a higher stiffness than flexible catenary.
Does this statement mean the following definitions?
The definition of the flexible catenary is an extensional component (bending is negligible). The definition of the rigid catenary is a bending component (axial extension is negligible).
Please add the definitions of "flexible catenary" and "rigid catenary" to the introduction. This journal covers various engineering topics. Therefore, most readers do not know the jargon language.
Major Revision 3
Based on those methods, the authors carried out a set of experiments to conclude that the values in the paper are the most appropriate. These values are not published but they have used successfully when comparing simulated results and real results.
Please add the author’s experimental result to the manuscript. This manuscript is purely numerical work that does not have any comparisons with the experiment or references. Therefore, the experimental result is necessary to improve the reliability of the manuscript. There is no need to hide the experimental result if the authors have them. Adding the experimental result to appendix is sufficient if changing the manuscript structure is difficult.
Minor Revision 1
Line 10: the authors present a new mathematical
Please remove "new." The combination of exiting methods is not new/novel. If we admit that the combination of exiting methods is new/novel, millions of peer-review papers could be published. We must avoid the flooding of such papers.
Minor Revision 2
Line 67: we present a methodology based on a novel mathematical model
Please remove "novel" and replace this sentence with e.g. "we present a methodology based on the combination of well-known mathematical models."
Minor Revision 3
Line 769: a new model for the pantograph-rigid catenary interaction
Please remove "new."
Minor Revision 4
Line 772: In order to build this new model,
Please remove "new."
Author Response
"Please see the attachment."

Reviewer 2 Report
Unfortunately I cannot state any improvement at all; the last modifications introduced a number of typos (like "overlapping spam" in the abstract.) There are also several formulations like "This amount of studies involving elastic catenary is because the high performance ...", which need to be rewritten. My main critical remarks have not been addressed at all: The numerical treatment should be carefully checked by a expert for Numerical Analysis, the proposed methods and explanations sound very amateurish. In this regard there are two main critical points: The equations of motion (18) are differential-algebraic equations of index 3, which need a sophisticated treatment; the authors make a lot of fuss about the singularity of the mass matrix. (The statement, that for the inverse of a diagonal matrix only the reciprocals of the diagonal elements - if non-zero - need to be calculated, is really trivial. The problem with that kind of equations is much deeper.) The second problem with the model is the possible occurence of impacts: If after loss of contact the pantograph hits the stiff catenary, there might be impacts. I cannot find the treatment of that case. The presentation of the commercial (?) software package shouldn't occur in a scientific article, unless it constitutes a real innovation, like say the provision of LAPACK.Author Response
"Please see the attachment."

Reviewer 3 Report
I accept all the responses of the authors, and this paper deserves to be published with a good contribution to the community of pantograph-catenary interaction.
Author Response
"Please see the attachment."

This manuscript is a resubmission of an earlier submission. The following is a list of the peer review reports and author responses from that submission.
Round 1
Reviewer 1 Report
Summary of Manuscript
In this manuscript, a dynamic model for a pantograph-rigid catenary system is presented. First, numerical catenary and pantograph models are presented. Second, simulation parameters for the two models are described. Third, a contact force modeling is presented. Fourth, a numerical integration method is explained. The integrated numerical model is implemented in a simulation tool called RICATI. By using the tool, four simulations are performed.
Comments
The reviewer appreciate sharing the work. The reviewer’s opinion is major revision.
First of all, English writing in the manuscript is complicated and difficult to understand. Many of sentences are illogically long. Several sentences do not have a verb. The reviewer strongly recommends a significant improvement of the English writing. Commercial English editing service may be a good way to improve the English writing. If the improvement cannot be found in the revised manuscript, the reviewer would not recommend this manuscript for publication.
There are 8 major revisions and 13 minor revisions.
Major Revision 1
What is the research question/problem that the authors tackled? In Section 1, a phrase “this problem” appears several times. However, the definition of “this problem” is not clearly written. In the revised manuscript, please clearly state the research question/problem.
For example, “The problem that we solve in this paper is … This problem has not been solved by other researchers in the past. In this paper, however, our novel proposed method/idea/theory can solve this problem.”
Major Revision 2
Line 40: it also has a special relevance when the interaction occurs with rigid catenary as it happens in tunnels and underground roads, and where the scientific literature is minor.
What is the theoretical difficulty to handle a rigid catenary compared with a flexible catenary? The reviewer thinks that the rigid catenary modeling is much simpler than the flexible catenary modeling. Therefore, if the catenary is considered as a rigid body, the novelty in the modeling theory would become weak.
Major Revision 3
What is the theoretical novelty in the authors’ modeling? Pantograph modeling? Catenary modeling? Contact modeling? Numerical integration? The reviewer thinks that the modeling method in this manuscript is based on very classical spring mass, linear finite element, and Lagrange multipliers. Thus, it is difficult for the reviewer to find the theoretical novelty. The reviewer does not think that a rigid catenary modeling is novel, because it is just a simple (or high-stiffness) version of the flexible catenary modeling. In the revised manuscript, please clearly state the theoretical novelty. For example, “The novelty of the proposed modeling is …”
Major Revision 4
What is the advantage of the authors’ proposed modeling? Computational time? Stability of calculation? In the revised manuscript, please clearly state the advantage.
Major Revision 5
Mathematical letters are not unified in the manuscript. For example, yi is italic in Eq. (3), while yi is upright (non-italic) in Fig. 2 and the sentences. Please carefully unify all mathematical letters in the equations, sentences, and figures.
Major Revision 6
The figures do not help the reviewer understand the modeling and simulation results well. In Fig. 1, please show where the components in Figs. 2, 3, and 4 are located (or related to Fig. 1).
Major Revision 7
Line 98: 2.2 Beam model for rigid catenary
This subsection title is confusing. Is catenary a rigid body or a flexible beam?
Major Revision 8
Please modify Figs. 12-17.
Figures should have labels for the x- and y-axes.
The current fonts are unclear and small.
Separated figures should have each caption such as (a) Average value of pantograph effort (b) Standard deviation of pantograph effort (c) maximum value of pantograph effort, and (d) minimum value of pantograph effort.
Please add more information to the captions to Figs. 12-17. “Graphs of simulation 1” is too short and difficult to understand.
Minor Revision 1
Line 18: That encourage us -> That encourages us
Minor Revision 2
Please relate Fig. 1 with Eq. (1) by describing qc1, qc2, qp1, qp2 in Fig. 1.
Minor Revision 3
In Eq. (2), the subscripts “c” and “p” seem to represent “catenary” and “pantograph”. What does the subscript “n” represent?
Minor Revision 4
Line 121: Being α and β two constants that are determined from two assumed values of the critical damping for two different frequencies of interest. In our case for 1Hz a critical damping percentage of 0.5% was assumed, and for 15Hz of 1%.
The first sentence does not have a verb.
In the revised manuscript, please explain why 1 Hz and 15 Hz vibrations are chosen for the damping determination and why 0.5% and 1% damping are chosen.
Minor Revision 5
Figure 3: 110 -> 110 mm
Minor Revision 6
In Eq. (5), the left-hand side is mass m. Does the right-hand side E represent Young’s modulus? This equation is confusing and difficult to understand. In the revised manuscript, please explain Eq. (5) more.
Minor Revision 7
Line 152: Afte -> After
Minor Revision 8
Line 154: 4228994mm4 -> 4228994 mm4
Line 155: 5421787mm4 -> 5421787 mm4
These values have too many digits. Please consider significant digits.
Minor Revision 9
In Eq. (7), the mass matrix is rank-deficient. Thus, its inverse calculation is impossible. In the revised manuscript, please explain how the mass matrix is handled in the numerical integration.
Minor Revision 10
Line 206: dynamic equations 1 -> dynamic equation 1
Minor Revision 11
In Section 3, “meters”, “m.” and “m” are mixed. Please unify them.
Minor Revision 12
Please carefully follow a journal rule about a reference format. For example, Ref. [1] has 652-667, while Ref. [2] has Pp. 231-252.
Minor Revision 13
Please carefully check all reference information. For example, in Ref. [13], H.S. Sugiyama seems to be wrong. H. Sugiyama is correct.
Reviewer 2 Report
The article deals with a very common, but strongly simplified system. The applied solution method is quite simple, and in some aspects not well worked out.
For example the rigid catenary is modelled by finite elements; one would assume, that a rigid catenary can be properly described by some smooth curve. The section 2.8 about deforming the beam doesn't sound appropriate.
Moreover, the authors assume, that there can be some take-off. But when the pantograph touches the catenary again, one would expect some impact dynamics.
The implemented numerical scheme is quite simple, there could be some problems with stability and accuracy. I see no necessity to show details like the screen layout in a scientific article.
From the scientific point the article is very poor. The mechanical equations could be derived by undergraduate students. Equations with contact forces and constraints have now been solved successfully for decades, there is no new scientific content detectable in the manuscript.